# Design of Bifunctional Nanocatalysts Based on Zeolites for Biomass Processing

**DOI:** 10.3390/nano13162274

**Published:** 2023-08-08

**Authors:** Valentina G. Matveeva, Lyudmila M. Bronstein

**Affiliations:** 1Department of Biotechnology, Chemistry and Standardization, Tver State Technical University, 22 A. Nikitina St., 170026 Tver, Russia; matveeva@science.tver.ru; 2Regional Technological Centre, Tver State University, Zhelyabova St., 33, 170100 Tver, Russia; 3Department of Chemistry, Indiana University, 800 E. Kirkwood Av., Bloomington, IN 47405, USA

**Keywords:** bifunctional, catalysis, nanoparticle, zeolite, biomass

## Abstract

Bifunctional catalysts consisting of metal-containing nanoparticles (NPs) and zeolite supports have received considerable attention due to their excellent catalytic properties in numerous reactions, including direct (biomass is a substrate) and indirect (platform chemical is a substrate) biomass processing. In this short review, we discuss major approaches to the preparation of NPs in zeolites, concentrating on methods that allow for the best interplay (synergy) between metal and acid sites, which is normally achieved for small NPs well-distributed through zeolite. We focus on the modification of zeolites to provide structural integrity and controlled acidity, which can be accomplished by the incorporation of certain metal ions or elements. The other modification avenue is the adjustment of zeolite morphology, including the creation of numerous defects for the NP entrapment and designed hierarchical porosity for improved mass transfer. In this review, we also provide examples of synergy between metal and acid sites and emphasize that without density functional theory calculations, many assumptions about the interactions between active sites remain unvalidated. Finally, we describe the most interesting examples of direct and indirect biomass (waste) processing for the last five years.

## 1. Introduction

Biomass types include dedicated energy crops, forestry waste, agricultural crop residues, algae and other marine sources, wood processing residues, municipal waste, and wet waste such as sewage sludge from municipal wastewater, etc. [1]. An estimated biomass output is 100 billion metric tons per year. Three major ways of biomass processing are (i) thermo-chemical conversion of biomass (combustion, pyrolysis, gasification, liquefaction), (ii) biochemical conversion of biomass (anaerobic digestion and fermentation), and (iii) physicochemical conversion of biomass (including fractionation and depolymerization of biomass) [1,2].

Biomass, and especially biomass waste of different origins, are excellent sources for the environmentally sustainable fabrication of various nanomaterials, such as biochars [3,4,5,6], nanocellulose [7,8,9,10,11,12,13], porous carbons [5,14,15,16,17,18], carbon nanofibers [19,20], carbon quantum dots [21,22,23,24,25,26,27,28,29], graphene [28,29,30], lignin-based materials [31,32,33,34,35,36,37,38], etc. The main methods of obtaining such materials from biomass include thermal, microwave and ultrasound treatments [39], as well as pyrolysis carbonization for carbon-based materials [40,41]. In these cases, biomass processing also includes activation before and/or after thermal treatment with acids or bases as well as catalytic reactions with metal-containing catalysts [42]. The other important group of products that can be acquired from biomass is biofuels and value-added chemicals that could be obtained via catalytic reactions from biomass [43,44,45,46,47,48,49,50,51,52]. For years heterogenous catalysts based on different metals and supports have been involved in these reactions [53,54,55,56,57,58,59,60,61,62,63,64].

Among heterogeneous catalysts, bifunctional ones consisting of mono- or bimetallic nanoparticles (NPs) and zeolite supports have received considerable attention due to excellent catalytic properties in many reactions, including biomass processing, tandem reactions, etc. [65,66,67,68,69,70,71,72,73]. The term “Bifunctional catalysts” means that two types of catalytic sites are present in a single catalyst: metal sites from NPs and acid sites from the support. This allows carrying out simultaneous or tandem reactions, changing a reaction pathway due to dual catalytic action or changing the target products. All these advantages would be impossible with traditional metal-containing heterogeneous catalysts.

For the last ten years, a number of reviews have been published that were focused on the bifunctional catalyst structure and/or the reactions relevant to biomass processing [73,74,75,76,77,78,79,80,81,82]. At the same time, we believe that two crucial aspects of the NP/zeolite catalyst structure and function have been underrepresented, such as the zeolite/catalyst modification to improve catalyst integrity and catalytic performance as well as interactions between metal and acid sites that are normally critical for the success of the catalysts. Considering that the number of publications in this field has doubled from 2018 to present compared to the previous five years, in this short review we will analyze the literature published from 2018 through March 2023 with a major focus on the synergy or lack thereof between metal NPs and acid sites of zeolites as well as their controlled modification. To allow better understanding of interactions of catalytic sites in bifunctional catalysts, first, we will describe the major types of zeolites used in catalysts, as well as the fabrication methods of bifunctional catalysts. In the following sections, we will analyze (i) the modification of zeolites and catalysts, focusing on the structural integrity, morphology, porosity, and the control of the charges/acidity, (ii) interactions between active sites in bifunctional catalysts, and (iii) examples of direct and indirect biomass processing reactions. Here, direct biomass processing means that biomass or its waste is a substrate in the catalytic reaction, while in the indirect processing, the substrate is a platform chemical obtained from biomass. This review structure helps us to identify the most important developments in the field as well as the prospects for further research.

## 2. Types of Zeolites

Zeolites are aluminosilicates that are characterized by well-defined porosity and crystallinity [83,84]. The porous structure contains either cages or channels (or both) of different sizes (Figure 1) [85]. The presence of Al in zeolites provides negative charges, which result in acidic protons or cations for neutrality. The higher fraction of Al leads to higher acidity; thus, the acidity of zeolites can be controlled by the Al/Si ratio or by the incorporation of other species (for example, Ti, La, etc.). As is illustrated in Figure 1, the three major zeolite types are MFI (for example, ZSM-5, HZSM-5), FAU (for example, Y), and BEA (β). The other representatives of MFI-type zeolites are fully silicious silicalite-1 (S-1) and Ti-containing TS-1, including tetrahedral units of SiO_4_ and TiO_4_ [86]. MFI-type zeolites possess two interconnected channel systems containing pentasyl units and are considered medium pore-size zeolites [87,88]. FAU-type zeolites contain larger pores [89], while the BAU-type zeolite porosity is in between those of MFI and FAU (Figure 1) [90]. Diverse types and sizes of pores allow for various molecules either to penetrate the support or to be filtered out, depending on their size.

Normally, zeolite acidity is originated from Bronsted acid sites directly attributed to Al atoms in a tetrahedral [AlO_4_] framework [84,91,92]. At the same time, zeolites include substantial amounts of Lewis acid sites resulting from dehydration of the Bronsted sites or other sources. The important parameter is the ratio between Bronsted and Lewis acid sites, which is also dependent on the Si/Al ratio. It decreases at a low Si/Al content. It is worth noting that zeolites can also be basic when they are subjected to ion exchange with alkali cations or when they contain inclusions of basic metals.

It is worth noting that besides the major types of zeolites discussed above, new modifications were introduced, including layer-like hierarchical [93,94,95,96] and mesoporous zeolites [97,98,99,100]. A recent example shows a remarkably different outcome of the catalytic reaction, depending on the zeolite type, including porosity. In tandem reactions, it is very important to achieve a high selectivity for each reaction. Cho et al. discovered that an adjustment of the pore size in bifunctional catalysts based on Pt NPs and zeolite plays a crucial role in achieving high selectivity [101]. The encapsulation of Pt NPs in H-BEA zeolites allows for one-pot transformation of cyclopentanone (CPO) (obtained from biomass) to C_10_ cyclic hydrocarbons (bicyclopentane and decalin) with a total yield reaching 78%, a remarkable accomplishment. In the case of MFI, whose pores are smaller than those of H-BEA, mainly cyclopentane (~70%) is formed. These data show size selectivity of zeolite micropores towards bulky reactive intermediates.

## 3. Methods of the NP Formation in Zeolites

The key methods for metal-containing NP formation in zeolites include (i) wet impregnation of premade zeolites [102,103,104,105,106,107,108], (ii) physical mixing of metal compounds/NPs and zeolites [109,110,111,112], (iii) ion exchange in zeolites [113,114], and (iv) encapsulation of NPs in zeolites normally during a simultaneous formation of both constituents of the nanomaterial [103,115,116,117]. For each example, we will indicate a particular reaction in which the catalyst was utilized, and we will indicate the correlations (if any) between NP/catalyst characteristics and the catalytic process.

### 3.1. Impregnation

Wet impregnation is most frequently used because of its simplicity and the possibility to employ commercially available zeolites. Normally, wet impregnation is followed by reduction or calcination. For example, ZSM-5-based catalysts with different Al/Si ratios and containing NiO NPs (10–20 nm in diameter) were prepared by impregnation and tested in hydrodeoxygenation, decarboxylation and hydrocracking of palmitic acid [118]. It was determined that Ni loading influenced the ratio of Lewis/Brønsted acids, which in turn controlled the reaction outcome. In the case of bimetallic Ni/Fe NPs, sequential impregnation followed by calcination has been employed [102]. Here, the Ni/Fe-BEA catalyst showed much higher selectivity in the hydropyrolysis of eucalyptus leaves to polyaromatic hydrocarbons compared to monometallic Ni-BEA.

A major shortcoming of this approach is a tendency to the formation of large NPs with a broad NP size distribution, which diminishes interactions with zeolite acid sites. It is noteworthy that it is not always the case. In a remarkable development, the authors reported RuW alloy NPs (1–4 nm) prepared by wet impregnation on HY zeolite with a high silica content for the in situ processing of lignin (least valuable part of lignocellulosic biomass) [106]. This resulted solely in the formation of benzene due to the combination of Brønsted acid catalyzed conversion of sp^2^ to sp^3^ bonds of lignin. At the same time, RuW species facilitated the hydrogenolysis of the C-O bonds with hydrogen extracted from lignin. Even more controlled NP formation via impregnation was reported when very small Ru NPs (~1 nm in diameter) were formed in the zeolite Y micropores (Figure 2) [119]. Apparently, small NPs can be obtained by impregnation if the diffusion of the precursor metal compound is hindered by small pores, defects, traps, etc.

Impregnation in hydrothermal conditions was employed to place Pd NPs (with sizes below 3.5 nm) on nanocomposite containing carbon with numerous defects on the surface of HZSM-5 zeolite spheres [120]. These defects serve for catching the Pd NPs, limiting their size and promoting substrate adsorption. This leads to the successful hydrodeoxygenation of vanillin to [2-methoxy-4-methylphenol (a favorable liquid biofuel) with high selectivity. The authors believe that both the high concentration of defects on the nanocomposite surface and the high dispersion of Pd NPs play a crucial role in the success of this reaction.

### 3.2. Mixing

Mixing is the simplest method of bifunctional catalyst formation, but it rarely results in well-defined catalytic structures and in optimized catalytic processes. Co NPs incorporated in mesoporous Y zeolites were prepared by a melt infiltration method after mixing and studied in the direct conversion of syngas (CO/H_2_) into three different biofuels: diesel, gasoline, and jet fuel [121]. The authors discovered that the outcome of the reaction (the product distribution) is determined by the porosity and acidity of the zeolite and can be tuned at will. The Co NP sizes were in the range of 14–16 nm. Cu/Zn bimetallic NPs with diameters of 6.2 ± 2.0 nm were mixed with a number of acidic supports, including zeolites ZSM-5 and Y, to catalyze a synthesis of methanol from syngas and the methanol transformation to dimethyl ether or hydrocarbons [111]. The authors stated that the metal and acid sites are in close proximity in these catalysts, although the mixing of solids and calcination was used as a preparation technique, which normally provides a rather crude distribution of species.

In another avenue, mixing was used to combine two prefabricated catalysts, one of which contains NPs and the other is zeolite. Such a bifunctional catalyst mixture was proposed by Arslan et al. [110]. The nanocomposite catalyst was prepared by mixing ZnCr_2_O_4_ NPs and H-ZSM-5 and tested in one-step transformation of syngas to an aromatic hydrocarbon. High selectivity for aromatics was provided by short straight channels in H-ZSM-5, which demonstrate low diffusion resistance for these molecules. The reaction was catalyzed by ZnCr_2_O_4_ NPs, while the function of acid sites of zeolite was not realized in this study.

An interesting example of the catalyst mixture was discussed in Ref. [70]. Here, the catalyst mixture referred to as a tandem catalyst was prepared by mixing Pd/ZnO, which converts CO_2_ to methanol, and biomass-derived ZSM-5, which catalyzes methanol dehydration to dimethyl ether (DME)—a high value chemical. The authors compared the above tandem catalyst with the bifunctional catalyst (Pd/ZSM-5) and found that the close proximity of the catalytic sites in the latter was detrimental to the DME production (selectivity < 3%) due to ion exchange, while for the tandem catalyst, the selectivity to DME reached 31%. This work demonstrates that on a rare occasion, close proximity of active sites in a bifunctional catalyst can be disadvantageous.

### 3.3. Ion Exchange

An ion exchange method is infrequently used for bifunctional catalyst synthesis as ion exchange is often slow and results in low metal ion loading. The major advantage of this method is small, well distributed NPs. The bifunctional catalysts with well-dispersed Ni NPs (3.5 or 6.1 nm depending on the precursor loading) in the BEA zeolite were fabricated by ion-exchange–deposition–precipitation and tested in a hydrodeoxygenation of guaiacol towards hydrocarbons [113]. The catalysts showed much better catalytic performance than those prepared by wet impregnation with larger Ni NPs. It is worth noting that the conversion of guaiacol did not depend on the catalyst preparation method, while the product selectivity was strongly affected. 

A bifunctional catalyst consisting of small Ru NPs well-dispersed on H-β zeolites were employed in the transformation of furfural (FAL) into 3-acetyl-1-propanol (3-AP) [114]. The authors prepared the catalysts using either ion-exchange or impregnation. In the former case, 1.3 nm NPs with a narrow size distribution were obtained, while in the latter case, 2.5 nm NPs with a broad particle size distribution were formed (Figure 3). Smaller Ru NPs were found to provide much better catalytic properties, which were assigned to two factors: (i) more Ru active sites on smaller NPs and (ii) a better interaction between acid and metallic sites.

### 3.4. Encapsulation

Encapsulation is a complex but very sophisticated way for syntheses of bifunctional catalysts. Encapsulation can be carried out either by crystallization of zeolite around premade NPs (or complexes) or by the formation of both components at the same time or in a sequence. Wang et al. developed two different approaches for Co NPs containing HZSM zeolites, which dramatically altered the final product in the reaction of γ-valerolactone (GVL) obtained from lignocellulose [103]. In one approach (Co@HZ5), Co NPs were formed in silica, which was further converted into crystalline zeolite. In the other approach (Co/HZ5), Co NPs were formed by impregnation of zeolite, followed by calcination. As a result, for Co@HZ5, valeric biofuel was obtained, while for Co/HZ5, the product was pentane biofuel, demonstrating a complete switch of selectivity caused by different interactions of Co NPs with zeolite. For well-dispersed Co NPs encapsulated within HZ5 crystals (Co@HZ5), the synergy in the catalytic interactions was at its best, leading to GVL upgrading. In the case of Cu NPs, the same strategy was employed, where Cu NPs were encapsulated in highly crystalline zeolites producing valeric biofuel with high hydro-conversion efficiency [115].

One of the important reactions of biomass valorization is the selective dehydrogenation of ethanol to acetaldehyde. Small (~1.8 nm) Cu nanoparticles were encapsulated inside zeolites by an in situ approach first coordinating Cu ions with polyethylene-polyamine to prevent the Cu species precipitation during the zeolite formation [116]. In this way Cu NPs are located within zeolite cavities due to the interaction between zeolite and the metal complex. This also allows a significant fraction of the active (but normally unstable) Cu^+^ species. The encapsulated catalyst provided enhanced selectivity, activity, and stability vs. the non-encapsulated catalyst formed by wet impregnation.

The advantages of the encapsulation of metal NPs in zeolites vs. impregnation by metal compounds (resulting in limited catalyst stability due to agglomeration of NPs) were also demonstrated by Fu et al. in the development of the bifunctional Pd@HZSM-5 catalyst [71]. Here, Pd NPs were captured in HZSM-5 zeolites in situ and demonstrated excellent stability and high 3-acetyl-1-propanol yield in the conversion of 2-methylfuran. It is worth noting that this reaction is an important step in the processing of biomass-derived platform chemicals, such as furan derivatives, to value-added products.

Using an in situ encapsulation method, highly dispersed 1.8 nm Cu NPs were formed within TS-1 (titanium silicalite-1) zeolite (Figure 4) [117]. An exchange of protons for Na ions allows stabilization of the zeolite structure and excellent performance in the selective hydrogenation of biobased FAL to furfuryl alcohol (FOL). The authors discovered that restricting the zeolite environment for NPs promotes electronic interactions of Cu NPs and Ti species in Na-Cu@TS-1. This results in the inhibition of Cu NP aggregation and leaching, while Na species modify acidity and suppress side reactions.

The catalyst based on Ni NPs (2–5 nm in diameter) encapsulated in HZSM-5 has been utilized for the hydrodeoxygenation of palmitic acid to diesel-like biofuels [122]. The full deoxygenation of palmitic acid to C_15_-C_16_ alkanes with 100% selectivity using Ni-HZSM-5 was attributed to comparatively small Ni NPs well dispersed in zeolite. Again, a comparison with the catalyst prepared by impregnation shows a clear advantage of the encapsulated NPs.

The above discussion demonstrates that most well-defined and efficient catalysts are prepared by encapsulation due to limiting NP growth, inhibition of NP aggregation and metal leaching. However, from a technological point of view, these preparations are often too complex and time and labor consuming. The formation of small NPs in bifunctional catalysts allowing high catalyst efficiency can be achieved by simple impregnation if the zeolite support contains defects or folds limiting the NP growth. The values of NP sizes prepared by different methods are presented in Table 1.

## 4. Zeolite/Catalyst Modification for the Enhancement of Bifunctional Catalysts

A modification of zeolites/catalysts is carried out for the strengthening of the zeolite framework, a change in acidity, dealumination, etc. It can be performed with the preformed zeolite, with the bifunctional catalyst or at the stage of the zeolite formation.

### 4.1. Influence of Porosity on Mass Transfer and Catalytic Processes

Zeolite porosity is a part of the morphology modification introduced at the stage of the zeolite synthesis. Small pore (microporous) zeolites of different types possess valuable properties and find numerous applications in catalysis [123]. Even among those, it was reported that zeolites with larger pores (BEA) showed better conversion from sorbitol to isosorbide than MOR (mordenite) and MFI with smaller pores [124]. It is noteworthy, however, that these zeolites also possess different acidity so the pore influence is not quite straightforward.

In many publications, the importance of the creation of mesopores in zeolites and/or hierarchical porosity (including meso- and macropores, along with micropores) is discussed from the viewpoint of optimizing mass transfer [81,125,126,127,128]. The study of mesoporous β zeolites in the catalytic benzylation of naphthalene showed that the increase in the amount of mesopores improved the diffusion of the reactants and products and facilitated access to acid sites, resulting in improved catalyst performance [129].

A catalytic behavior of MFI nanosheets with hierarchical porosity and the microporous parent MFI zeolite was studied in the methanol-to-propylene reaction [130]. It was demonstrated that the former displays much better performance due to the shortening of the diffusion path length, thus reducing mass transport limitations. MFI with a microporous carcass embedded in the ordered mesoporous structure was loaded with Pt NPs and tested in the hydroconversion of *n*-decane [131]. This test showed the formation of 10 membered ring products due to the size selectivity of the mesopores as well as the products typical for microporous MFI.

A bifunctional catalyst based on ultra-small Pt NPs formed on a hierarchical zeolite showed exceptional performance in the consecutive mild hydrodeoxygenation of 4-propylphenol to a nearly 100% cycloalkane product, significantly exceeding the performance of the catalyst based on conventional zeolite [132].

### 4.2. Modification with Ions

La ions have been utilized for the stabilization of the zeolite against deconstruction. He et al. suggested such a modification of the bifunctional catalysts based on Ni NPs and zeolite by incorporation of La ions into H-Y [72]. This significantly increased the stability of the catalyst performance in the liquid phase, which was assigned to decreasing coke formation, dealumination and metal NP aggregation. This strategy was employed by the same group for the fabrication of a bifunctional catalyst based on very small Ru NPs (~1 nm in diameter) formed by an impregnation method in the zeolite Y micropores, allowing for an active site “intimacy” (close proximity of Ru species to zeolite acidic sites) [119]. An additional modification of zeolite with La stabilizes the Y structure during catalysis by stopping lattice deconstruction and preserving the proximity of the catalytic sites. Such a catalyst significantly enhances selectivity towards pentanoic biofuels in one-pot hydrodeoxygenation of biomass-derived ethyl levulinate.

Sn ions have been introduced in the *β* zeolite during its formation for further fabrication of a hybrid multifunctional catalyst, Au/CuO-Sn-*β*. It has been prepared by combining Au/CuO NPs and Sn-modified *β* zeolite followed by calcination [133]. The addition of Sn led to Lewis acid sites, which, in turn, improved the catalytic transformation of biomass-derived glycerol to methyl lactate—a monomer for a biodegradable polymer.

Intimate metal–acid interfaces were probed in the bifunctional catalyst based on Pd NPs encapsulated in the Y zeolite and modified with sulfonic acid groups [134]. This approach immobilizes metal and acid sites in close proximity due to the rigidity of the zeolite crystal. Moreover, because sulfonic acid groups are much stronger than the acid sites of zeolite, the success of the hydrodeoxygenation of FAL (a multistep reaction) is attributed to the interfaces of the former with metal NPs.

Zeolite 5A (with cage-like 0.5 nm pores) was impregnated with trifluoromethanesulfonic acid (to introduce strong acid sites) without solvent and after drying was used for the thermal deposition of Ni NPs [135]. The catalyst was employed for the processing of model compounds imitating lignin (oxybis(methylene)dibenzene and benzyloxybenzene). It was determined that Ni NPs are very large (under 100 nm) and located on the surface of zeolites. Nevertheless, this catalyst allows the production of protons and their transfer, leading to hydrogen formation, i.e., catalytic hydroconversion. However, in our opinion, the morphology of this catalyst is so ill-defined, it should not be recommended for any further studies.

### 4.3. Dealumination, Desilication

Dealumination normally decreases the amount and strength of acid sites, while desilication creates the opposite effect. Both actions can significantly modify the catalyst properties. It is noteworthy that some side reactions are suppressed at high acidity, while the other side reactions are blocked at low acidity or at different types of acid sites. This determines the choice of zeolite and its modification. A detailed study of the reaction routes for the transformation of methyl palmitate to jet biofuel was presented in ref. [136]. The authors used Ni NPs in the desilicated (by the NaOH treatment) Y zeolite. Both calculations and catalytic tests allowed the authors to determine the most probable mechanism. It was shown that desilication increases the zeolite acidity, suppressing side reactions. Quantum chemistry calculations revealed that with such catalysts, the hydrodecarboxylation reaction is more probable than hydrogenolysis and decarboxylation.

An extreme case of dealumination is fully silicious zeolite, silicalite-1, which contains no Al and is characterized by very weak Brønsted acid sites. It has been utilized for the fixation of Pd NPs either by impregnation or encapsulation (Figure 5) [137]. The modulation of the support wettability by the functionalization of silanol groups resulted in the altered diffusion of reacting molecules and exceptional activity and selectivity in the hydrogenation of FAL to furan.

### 4.4. Incorporation of Fluorine

A promotional effect of F on bifunctional Pd/HZSM-5 catalysts was reported by Jiang et al. [138]. Here, fluorine, which is introduced as NH_4_F together with the Pd precursor, replaces OH group in zeolite, forming the F-Al bond. This alters the acidity, hydrophobicity, and surface morphology of the catalyst. A variation in the fluorine amount allows one to control the above properties. The fluorine modification improved the catalytic properties in the selective hydrodeoxygenation of ketones obtained from biomass. In another example of modification with F, Ru NPs in MFI zeolites were prepared by impregnation using a novel method for the zeolite synthesis [139]. Instead of crystal seeds or solvent, the authors utilized fluorine-containing species for ZSM-5 crystallization. The coordination of fluoride ions with Al^3+^ ions leads to six coordinated ‘F-Al-O-Si’ species that promote the growth of tetrahedral [AlO_4_] fragments in the zeolite. These bifunctional catalysts were employed in the successful hydrogenation of levulinic acid and glucose.

### 4.5. Post-Fabrication Modification

In Ref. [140], a multicomponent catalyst synthesized by forming Pt NPs in the dealuminated *β* zeolite was coated with Mg(OH)_2_ and utilized to obtain 1,2-propanediol from biomass-based sucrose. The coating with Mg(OH)_2_ resulted in the weakening of the Lewis acid sites as well as an appearance of weak and strong alkaline sites. Altogether, this resulted in an enhanced catalytic performance compared to the catalyst without Mg(OH)_2_ due to suppression of side reactions.

A combination of ZrO_2_ NPs with ZSM-5 by ball-milling resulted in the catalysts converting methyl levulinate into GVL [112]. Because selectivity to GVL was decreased due to a side reaction on the Lewis and Brønsted acid sites, such bases as pyridine and 2,6-dimethylpyridine have been utilized to completely suppress side reactions, improving the GVL outcome.

### 4.6. Modification with Magnetic NPs

Magnetically recoverable catalysts have been at the forefront of the fabrication of novel catalysts due to easy magnetic separation, allowing one to conserve energy and materials and to simplify both catalyst purification and reuse in catalytic processes [141,142,143,144,145,146,147,148,149,150,151,152,153,154,155,156,157,158]. Among bifunctional catalysts based on zeolites, there are several examples where magnetic NPs were added to ensure facile magnetic separation [159,160,161,162,163,164,165,166,167]. At the same time, we found only a single example of such a catalyst utilized in biomass-related catalytic reactions. Prech et al. developed magnetically recoverable catalysts with the incorporation of iron NPs coated with carbon into the Y zeolite-bearing Lewis acid sites [140]. The catalyst was utilized in the hydrolysis of the marine-based polysaccharide, where acid sites of zeolites were catalytic sites, while magnetic NPs provided magnetic separation. Despite the fact that there are two functions here from the same nanocomposite, magnetic and catalytic, the magnetic function is irrelevant to catalysis.

### 4.7. Morphology Modification

One of the methods to tune zeolite ZSM-5 porosity and acidity is its formation around carbon NPs [168]. The high concentration of -C-O-C- groups on the carbon NP surface results in enhanced hydrophilicity during the zeolite formation, leading to hierarchical porosity and improved Brønsted acidity—important parameters for successful bio-refining.

Meeting the challenges of the formation of small metal NPs in mesoporous zeolites, the authors of Ref. [169] proposed steam-assisted recrystallization, creating an unusual shell-like morphology that stabilizes small metal NPs in microporous channels (Figure 6). To demonstrate exceptional morphological stability of Ni NPs as well as their excellent catalytic performance, methanation of CO_2_ was used as a test reaction. Despite a high reaction temperature (450 °C), no NP agglomeration was observed.

Xu et al. proposed an interesting morphology for the bifunctional catalyst to maximize metal–acid synergetic interactions [170]. For this, the authors developed a mesoporous core–shell catalyst with a ZSM-5 core and a shell formed by Pd NPs on Al_2_O_3_ for the hydrodeoxygenation of biomass-based compounds. For comparison, they also used a mixture of zeolite and Pd/Al_2_O_3_ and synthesized Pd NPs/ZSM-5 by impregnation and encapsulation. It is worth noting that encapsulation normally creates the greatest proximity between metal and acid sites. Nevertheless, the core–shell zeolite@Pd/Al_2_O_3_ catalyst showed the best activity, selectivity, and stability upon reuse compared to other catalysts. This was assigned to the highest metal–acid synergy, also suppressing the coke formation in the hydrodeoxygenation process, allowing for successful recycling. This explanation would be valid if the Pd/Al_2_O_3_ shell were thin, but it is not the case. It is noteworthy that Pd NPs contain both Pd^0^ and Pd^2+^ species, with the highest fraction of Pd^0^ in Pd/Al_2_O_3_, the highest fraction of Pd^2+^ in Pd/ZSM-5, and an intermediate amount of Pd-oxidized species in the core–shell catalyst. One might assume that this is the cause of exceptional catalytic properties. 

The cross-shaped (containing spherical and cuboid shapes) HZSM-5 zeolite was obtained with the help of seeds and piperidine as a structure-directing agent. It was filled with Ru NPs prepared by wet impregnation and tested in a tandem reaction of hydrogenolysis of guaiacol to benzene [171]. A comparison of the catalytic performance of the above catalyst with those based on the single morphology zeolites (either cuboid or spherical) showed that a cross-shaped morphology led to better catalytic properties in both catalytic steps. This was attributed to smaller and better dispersed Ru NPs in cross-shaped zeolite and better guaiacol adsorption that, in turn, was assigned to a high concentration of Lewis acid sites.

Thus, there are different aspects of the catalyst structure and properties that can be successfully influenced by the different modification approaches. Although technologically sound, a modification with ions (La) to prevent deconstruction, dealumination/desilication to control acidity, and morphology modification to control NP formation appear the most promising avenues.

## 5. Interactions between Zeolite Acid Sites and NPs

The major advantage of bifunctional NP-zeolite catalysts is the opportunity of acid sites of zeolites to “talk” to active sites of catalytic NPs and vice versa to weaken or to strengthen catalytic effects, allowing for high selectivity and activity even in tandem or simultaneous complex catalytic reactions.

A rearrangement of biomass-derived FAL to CPO via FOL is an important path for biomass valorization. Gao et al. synthesized Pd NPs in H–ZSM–5 zeolites using wet impregnation [105]. The catalyst allowed 98% selectivity toward CPO with the 120 h^−1^ specific reaction rate. Brønsted acid sites in zeolite facilitated the hydrogenation of FAL to FOL. At the same time, Lewis acid sites of zeolite allowed the hydrogenative rearrangement of FOL to CPO. The important finding was that bare H-ZSM-5 (no Pd) was not efficient compared to the Pd-containing catalyst, revealing the synergy between Pd species and acidic sites. 

*α*-Pinene is an important biomass-derived chemical that can be converted into pinane—an intermediate in preparation of fragrances and pharmaceuticals. However, *α*-pinene hydrogenation normally results in the mixture of *cis*- and *trans*-products, with *trans*-pinane being undesirable. Fan et al. developed Ru NPs modified by Ni in the Hβ zeolite, allowing for 98% selectivity to *cis*-pinane at 100% conversion [172]. This exceptional performance was assigned to the influence of Ni, which controls the ratio of Brønsted and Lewis acids and adjusts the hydrogen spillover between Ru and zeolite as well as modifies adsorption sites, altogether validating the importance of multifunctionality in such catalysts.

Dai et al. studied the transformation of biomass-derived FOL to pentanediols using several bifunctional Cu/MFI catalysts [173]. One of them, containing a certain combination of Cu^0^ and Cu^+^ species as well as Brønsted acid sites, demonstrated excellent catalytic properties. Density functional theory (DFT) calculations showed that successful ring-opening and hydrogenation should be assigned to the synergy between the Cu species and Brønsted acid sites; the latter impact the ring-opening in FOL, while the adsorption of the FOL methyl group on the Cu^+^ species promotes hydrogenation (Figure 7).

Wetchasat et al. synthesized several bifunctional catalysts, including 1 nm Pt NPs dispersed on hierarchical zeolites sheets using encapsulation with ethylenediaminetetraacetic acid to provide high-dispersion Pt NPs in the catalyst [132]. Using a model reaction of hydrodeoxygenation of 4-propylphenol to cycloalkane, the authors demonstrated the advantages of the above catalyst compared to larger Pt NPs stabilized on conventional zeolite or silicate. They believe that the location of 1 nm Pt NPs in the vicinity of Brønsted acid sites could improve the transfer of an intermediate between catalytic sites, thus promoting the desired catalytic process.

W(Ni)-zeolites prepared by impregnation and based on different zeolite types were studied as catalysts in the production of bio-based aromatic compounds from model biomass tar [174]. The W content in zeolites was found to raise the number of Lewis acid sites and total acidity, which could relate to the increased catalytic activity. On the other hand, the W species decrease Brønsted acidity and some other factors, which could also be beneficial for catalytic activity. Unfortunately, the interactions and influences observed in this work are too complex to provide a clear picture of synergetic effects.

It is noteworthy that there are numerous claims about intimate, close, excellent, etc. interactions between metal and acid sites, but in some papers, these claims are not verified. Only a combination of DFT calculations and experimental studies, including thorough nanomaterial characterization, allows one to validate both the most probable mechanism of the catalytic reaction and interactions between metal and acid catalytic sites [175,176,177,178,179,180,181,182].

## 6. Biomass Processing Catalytic Reactions with Bifunctional Catalysts

As was discussed in the Introduction, three major processes used for biomass upgrading include (i) the thermo-chemical conversion of biomass, (ii) the biochemical conversion of biomass, and (iii) the physicochemical conversion of biomass. Many of the above processes are catalytic. Heterogeneous catalysis is necessary for the transformation of biomass to platform chemicals and their further reactions to biofuels and value-added chemicals for pharmaceutical, cosmetic, and chemical industries [1,2,183]. In this review, we limit the discussion to those processes that are carried out with bifunctional zeolite/NP catalysts. 

The reactions associated with biomass processing can be divided into two major groups: (i) the reactions of the direct biomass (waste) processing, and (ii) the transformations of platform chemicals obtained from biomass (indirect biomass processing). The former case is especially appealing from the viewpoint of green chemistry; however, it is often very complex. Hence, the examples of direct conversion of lignocellulosic biomass are comparatively rare. In this section, we will discuss some recent examples of both types of catalytic biomass processing.

### 6.1. Direct Processing of Biomass

Protonated HBEA (*β*) zeolite containing mesopores in the range of 8–11 nm and Ni or Ni/Fe NPs demonstrated a considerable efficiency in the catalytic hydropyrolysis of eucalyptus leaves to aromatic monomers and biofuels [102]. Moreover, both types of catalysts display excellent selectivity to polyaromatic phenols via partial hydrogenation as well as ring-opening of polyaromatic hydrocarbons to aromatic compounds. Considering that Ni/BEA and Ni/Fe-BEA exhibited an analogous acid site concentration, the weaker hydrogenation activity and improved hydrogenolysis of the bimetallic bifunctional catalyst is attributed to the presence of Fe along with Ni (Figure 8). Altogether, the observed enhancement of the multi-step catalytic reaction is assigned to the interaction of both metal species and dissociated protons of zeolites.

Waste tire pyrolytic oil was converted to fuels using catalysts whose zeolite supports were also obtained from waste [104]. Ni-W NPs in such a zeolite were obtained by the impregnation of corresponding precursors, followed by a thermal treatment. The authors explored two different zeolite structures and two different metal loadings, but no clear advantage was found for either catalyst.

Successful catalytic pyrolysis of lignocellulose biopolymers into bio-oil (via partial deoxygenation) has been carried out with highly dispersed ZrO_2_ NPs formed on the surface of nanocrystalline or hierarchical ZSM-5 [184]. Here, Zr species adjust ZSM-5 acidity via a decrease in strong acid sites in zeolites and a production of new Lewis acid sites attributed to ZrO_2_.

A multifunctional catalyst based on bimetallic Cu-Ru NPs and HZSM-5 was employed in the one-pot processing of woody biomass to cyclic ketones and aromatic monomers [185]. In this multifunctional catalyst, the zeolite moiety with a Si/Al ratio of 100 was responsible for needed acidity for the successive depolymerization, dehydration, and isomerization of cellulose, while bimetallic NPs synergistically catalyzed the hydrogenation, hydrogenolysis and stabilization of lignin-derived intermediates. The authors believe this catalyst type can be a versatile platform for one-pot biomass processing to valuable chemicals.

Chen et al. chose to combine biomass—pine sawdust—with plastics to increase hydrogen content for a better yield of biofuels [186]. This mixture was used in catalytic conversion with Pd/trap-HZSM-5 catalysts. The most efficient catalyst was obtained when the Pd^2+^ compound was self-reduced by traps in HZSM-5. The traps were fabricated by a hydrothermal treatment of zeolite at 700° due to dealumination [187]. Increasing the concentration of Pd species in the traps followed by the reduction leads to sinter-resistant Pd NPs. The authors of Ref. [186] demonstrated that in this case, the smallest Pd NP size (5.4 nm) and moderate acidity promote the biofuel formation.

To achieve direct processing of plant biomass to gaseous and liquid intermediates involving several high-temperature processes, thermally stable catalysts need to be developed. Hu et al. synthesized such a catalyst by growing ZSM-5 on SiC nanowires [188]. At the same time, the authors adjusted the pyrolysis process, resulting in the optimization of both catalyst development and process strategy, enhancing the lifetime of the catalyst and promoting catalytic properties.

### 6.2. Indirect Processing of Biomass

One of the valuable platform chemicals in plant-based biomass is 5-hydroxymethylfurfural (HMF), whose transformation to 2,5-furandicarboxylic acid (FDCA) may allow the substitution of petroleum-based terephthalic acid. Salakhum et al. formed well-dispersed Pt NPs in alkaline-substituted ZSM-5 NPs with a hierarchical pore system using wet impregnation [107]. The catalyst showed remarkable performance in the HMF-FDCA reaction in mild conditions with 100% conversion and 80% FDCA selectivity. The authors assign their success to synergy between Pt NPs and alkaline ZSM-5 in the multi-oxidation reaction, especially in the case of the Ca^+^ modified zeolite.

To prepare efficient bifunctional catalysts for a successful one-pot conversion of xylose to tetrahydrofurfuryl alcohol, the authors formed the Ru/Hβ catalyst [189]. Here, the impregnation was followed by reduction with hydrazine (to remove chloride ions) and then by oxidation to form RuO_2_ at the corners and edges of Ru NPs. The combination of Ru with RuO_2_ suppressed side reactions and allowed for the high selectivity to the target products (Figure 9).

The cleavage of *β*-1,4-glycosidic bonds in cellulose is the first and crucial step in lignocellulosic biomass processing. The catalyst based on Ir NPs and the HY zeolite prepared by impregnation was utilized in the *β*-1,4-glycosidic bond cleavage of cellobiose with excellent activity and selectivity (>99%) under visible light at mild temperatures [190]. Cellobiose hydrolysis in such conditions was assigned to synergy between the Ir NPs (transforms light to thermal energy) and the acid sites of the HY zeolite, which provide active sites.

Trimetallic zeolite-based catalysts, Cu-Ni-Zn/H-ZSM5, have been synthesized by a wet impregnation method and utilized for the one-pot conversion of bio-derived levulinic acid to 1,4-pentadiol, a value-added chemical [108]. In this case, Zn was used to control the Cu-Ni alloy NP size and to improve reducibility. The authors carried out a thorough analysis of the functions of all parts of the catalyst, identifying the role of Lewis and Brønsted acid sites as well as the Cu-Ni alloy sites in this complex process.

A hybrid method to biomass-based jet fuel production from 2,3-butanediol (2,3-BDO) was proposed by Adhikari et al. [191]. The authors first carried out the conversion of 2,3-BDO to C_3+_ olefins using Cu NPs on ZSM-5 (with 98% selectivity and ~97% conversion) and then performed oligomerization to C_3_-C_6_ olefins with Amberlyst-36. This paper did not study the interaction between Cu species and zeolite active sites, thus not allowing one to elucidate synergy, if any, in the first reaction step. In Ref. [192], the authors discussed the catalyst based on Ru NPs on the NaY zeolite and emphasized that the Ru NP size and high surface area of the hydrophilic support are crucial in H_2_ production from glycerol and ethylene glycol, but the zeolite acidity of the proposed catalyst is not mentioned.

A mixture of two major compounds of bio-oil pyrolysis, guaiacol and acetic acid, was catalyzed over a bifunctional catalyst containing Ni_2_P NPs in ZSM-5 [193]. The paper describes various intermolecular interactions during the hydrodeoxygenation of the mixtures at various reaction conditions. A control of the selectivity of hydroconversion of FAL (biomass upgrading) was realized by the encapsulation of sub-nanometer Pd NPs in several MFI zeolites [194]. Surprisingly, different products such as furan, FOL, and 1,5-pentanediol are obtained, depending on the support. The authors determined that the zeolite microenvironment influences FAL adsorption and hydrogen activation due to cooperation between Pd NPs and the acid sites of the zeolite.

An unusual phenomenon of self-activation was observed for Pt/NaY catalysts in the base-free oxidation of biomass-derived ethylene glycol to glycolic acid and then to polyglycolic acid, a valuable product [195]. The thorough physicochemical characterization shows that upon oxidation, zeolite dealumination takes place, which results in the shortening of the Si-OH bond and the special interactions of Pt NPs with gluconic acid, which enhances the catalyst activity by a factor of two. This effect is explained by the electron enrichment of Pt species in oxidation.

To the best of our knowledge, bifunctional NP/zeolite catalysts are currently studied in research laboratories. For industrial applications in biomass processing, NP/zeolite catalysts need to be more stable and efficient, the features which are especially crucial for a technology transfer [81]. In general, catalytic processes can be implemented in modern biorefineries and biomass-upgrading systems because catalysis allows one to vary selectivity for many products obtained from biomass. However, to make it happen, additional efforts are needed to increase the stability of catalytic processes and to provide the further advancement of robust, well-defined multifunctional catalysts [183].

## 7. Conclusions

Bifunctional catalysts based on metal-containing NPs and zeolites and combining metal and acid sites have been successfully utilized in numerous catalytic reactions of biomass processing. In the majority of cases, the best catalysts are obtained when NPs are small and when there are significant interactions between metal and acid sites. These results are achieved for NPs encapsulated by zeolites. Various encapsulated bifunctional catalysts and their advantages compared to the catalysts obtained by simple impregnation were repeatedly recognized. Recently, innovative methods to achieve small NPs intimately interacting with the microenvironment of zeolites have been developed when zeolite morphology was tweaked. For example, wet impregnation was used for the formation of small NPs in the traps developed in the zeolite support at high temperature. The other approach is placing NPs in micropores developed in hierarchical (mainly mesoporous) zeolites. In other words, if the NP growth is restricted due to the suitable morphology modification, simple wet impregnation can result in excellent bifunctional catalysts. Beside the control of NP growth, the hierarchical porosity of zeolites allows for an enhanced mass transfer of reagents, thus intensifying the catalytic reactions.

The literature of the last five years also emphasizes the importance of the modification of NP/zeolite catalysts to control acidity (by dealumination, desilication, incorporation of F, ions, etc.), porosity (by the choice of zeolite or by the fabrication of mesoporous and hierarchical zeolites), and structural integrity (by incorporation of certain ions).

In this review, we were mostly “enchanted” by the interactions of metal catalytic sites and acid sites in bifunctional NP/zeolite catalysts. It is “assumed” that these interactions are responsible for excellent catalytic properties in complex and/or tandem reactions. However, “assumed” is the key word here. Even extensive physicochemical characterization, including the assessment of the NP size and composition, porosity and acidity of the support, does not allow one to evaluate these interactions. On the other hand, DFT calculations combined with thorough experimental studies allow for the validation of both probable reaction mechanisms and the interplay between metal and acid sites in these catalysts.

So far, the most important results in biomass processing with NP/zeolite bifunctional catalysts are the following. In direct processes, catalytic hydropyrolysis of eucalyptus leaves to aromatic monomers and biofuels with catalysts based on *β* zeolite containing mesopores and Ni/Fe NPs is especially successful due to the interaction of both metal species and dissociated protons of zeolites. Also, a remarkable development was reported for lignin processing in the presence of the RuW/HY catalyst to form solely benzene. Among indirect processes, we believe the transformations of biomass-derived platform chemicals such as furan derivatives to value-added products are particularly important because they are sources of many valuable target molecules.

Despite many remarkable accomplishments in this field, we clearly see two major shortcomings or niches, whose filling/remediation could be especially beneficial for catalyst design and biomass processing. First, despite the fact that magnetically recoverable catalysts literally overwhelmed catalyst development elsewhere, there are a few examples of magnetic bifunctional NP/zeolite catalysts, and almost none are used in biomass processing. This is surprising considering that the incorporation of magnetic (iron oxide) NPs in other catalysts did not show any detrimental effects on catalytic processes. Moreover, altering acid sites of the support in the presence of iron oxide NPs can be beneficial.

The second avenue for the successful development of this field is an increased focus on the direct processing of biomass or biomass waste, when they serve as substrates in one-pot transformations to biofuels or value-added chemicals. We recognize that these are very complicated processes, and several reactions can occur sequentially or concurrently, but the development of multifunctional catalysts with bi- and trimetallic NPs in zeolites could be promising. This would be a very important advancement of the field with huge environmental benefits.

For the further development of biomass upgrading, novel methods of biomass processing and the enhancement of catalyst stability and efficiency are extremely important. Both factors are impossible without the development of novel multifunctional catalysts. Bioprocessing plants combining thermochemical, biotechnological, chemical, and physical processes as well as novel catalysts will allow one to improve the yield, productivity and purity of multiple target compounds obtained from biomass.

## Figures and Tables

**Figure 1 nanomaterials-13-02274-f001:**
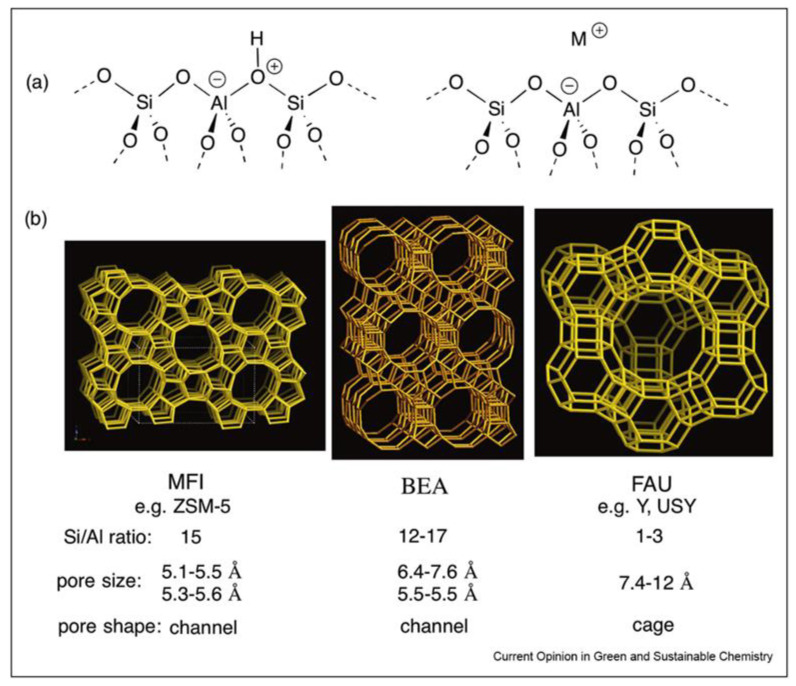
(**a**) The aluminosilicate framework of zeolites explaining their acidic properties; (**b**) structures, pore sizes and shapes of typical zeolites. Reproduced with permission from [85], Elsevier, 2018.

**Figure 2 nanomaterials-13-02274-f002:**
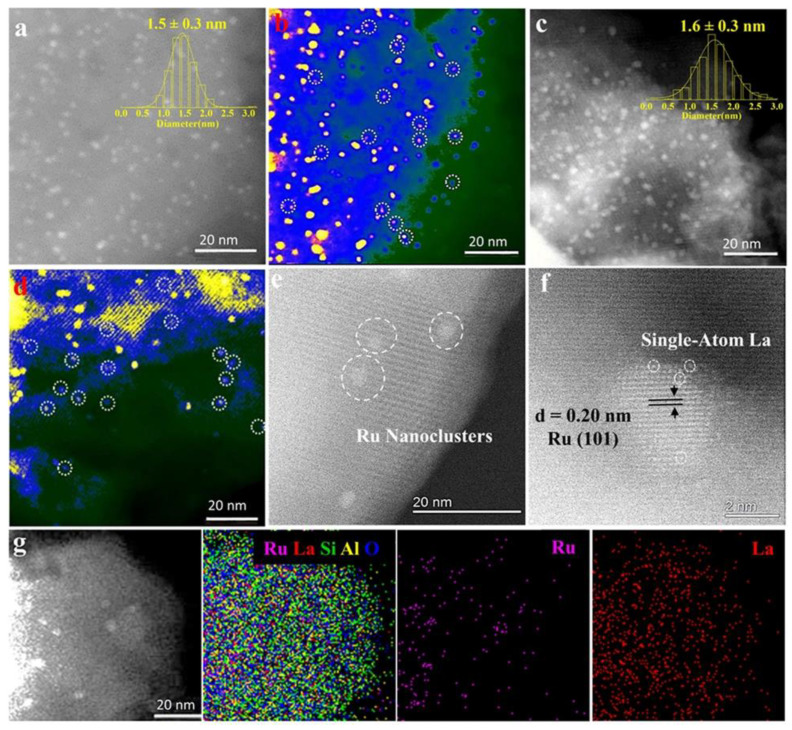
Structural characterization of the bifunctional catalyst materials under study. Representative aberration-corrected high-angle annular dark-field scanning transmission electron microscopy (AC-HAADF-STEM) images of (**a**,**b**) the Ru/H-Y and (**c**,**d**) the Ru/La-Y showing the existence of Ru nanoparticles confined in zeolite Y, corresponding particle size distribution derived from measurements of over 200 particles. (**e**,**f**) Atomic resolution of AC-HAADF-STEM images of the Ru/La-Y and (**g**) EDX spectral imaging of the Ru/La-Y and corresponding elemental maps: Ru pink, La red, Si green, Al yellow, and O blue, showing that Ru and La species are indeed highly dispersed in zeolite Y. Reproduced with permission from [119] Wiley, 2021.

**Figure 3 nanomaterials-13-02274-f003:**
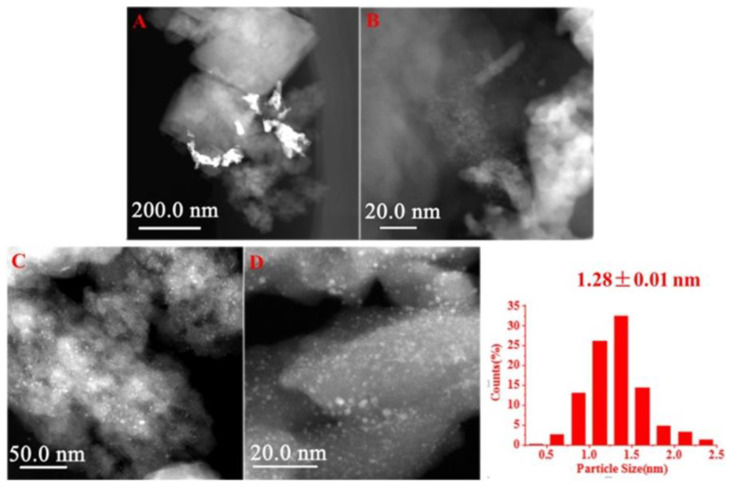
HAADF-STEM (high-angle annular dark-field scanning transmission electron microscopy) images of the catalysts: (**A**,**B**) Ru/H-beta-IM; (**C**,**D**) Ru/H-beta-IE-250 and particle distribution. “IM” stands for impregnation method, while “IE” stands for ion exchange. Reproduced with permission from [114], Elsevier, 2019.

**Figure 4 nanomaterials-13-02274-f004:**
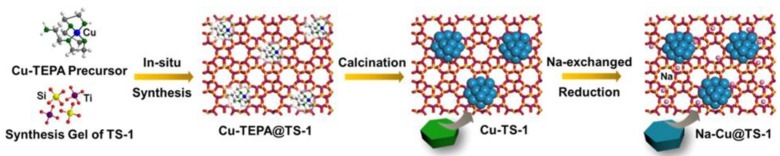
Schematic illustration of the in situ encapsulation approach for the synthesis of the Cu@TS-1 and Na–Cu@TS-1. Reproduced with permission from [117] the American Chemical Society, 2021.

**Figure 5 nanomaterials-13-02274-f005:**
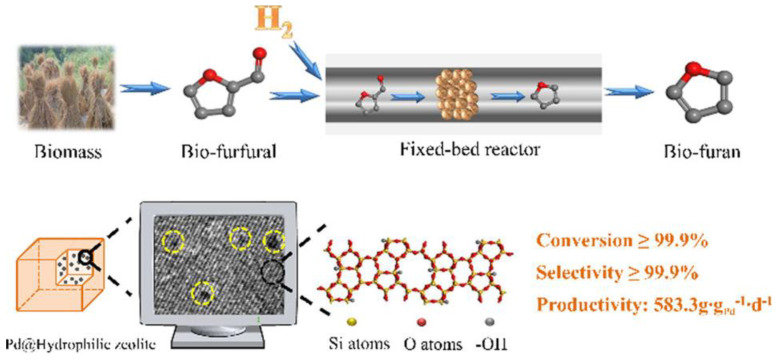
Catalytic strategy for producing bio-furan via selective hydrogenation of furfural. Reproduced with permission from [137], the American Chemical Society, 2018.

**Figure 6 nanomaterials-13-02274-f006:**
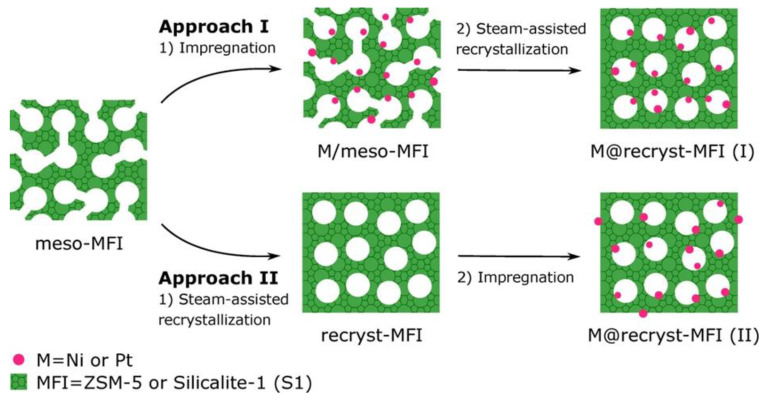
Schematic outline of a general method to encapsulate metal nanoparticles in zeolites. Reproduced with permission from [169], the American Chemical Society, 2019.

**Figure 7 nanomaterials-13-02274-f007:**
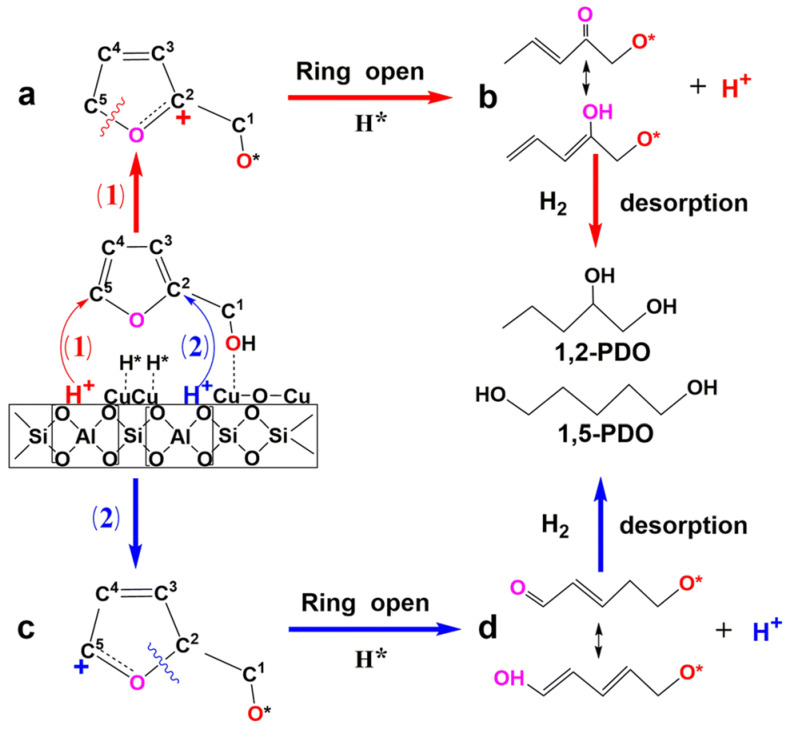
Possible pathways for the formation of 1,2-PDO (1,2-pentanediol) and 1,5-PDO (1,5-pentanediol) from furfuryl alcohol using copper-based zeolite catalysts. Reproduced with permission from [173], the Royal Society of Chemistry, 2022.

**Figure 8 nanomaterials-13-02274-f008:**
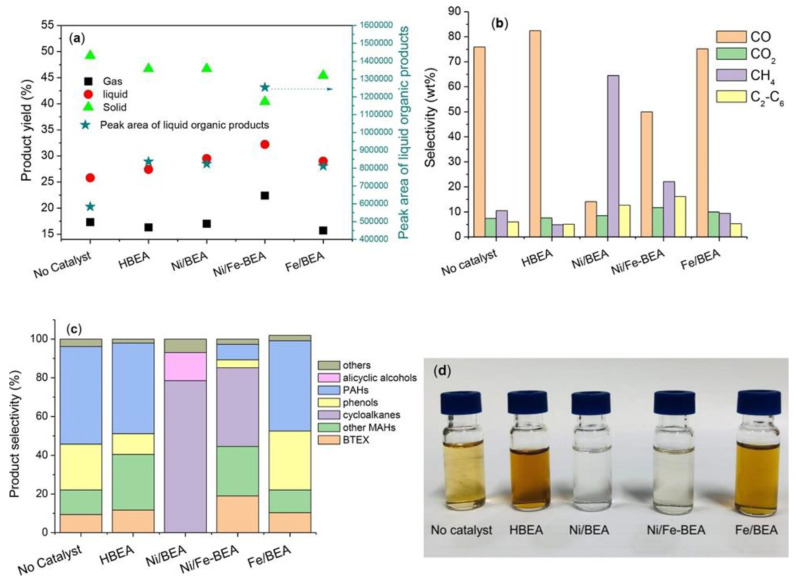
Product distribution (**a**), selectivity of (**b**) gas products and (**c**) liquid products from catalytic pyrolysis of eucalyptus leaves over different catalysts; (**d**) collected liquid products diluted in 20 mL dichloromethane. Other MAHs refer to monoaromatic hydrocarbons except for BTEX chemicals (benzene, toluene, ethylbenzene, and xylene). Reaction conditions: initial P_H2_ = 3.0 MPa, 350 °C, 0.5 g biomass, 0.2 g catalyst, 2 h. Reproduced with permission from [102], Elsevier, 2023.

**Figure 9 nanomaterials-13-02274-f009:**
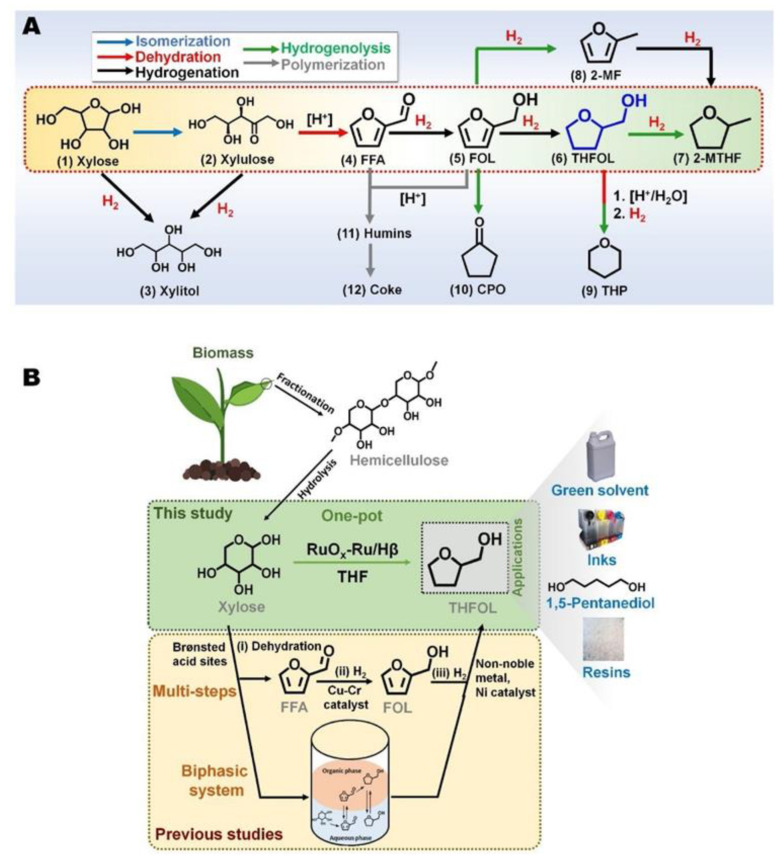
(**A**) Reaction pathways and byproducts for the conversion of xylose to tetrahydrofurfuryl alcohol (THFOL) and (**B**) biomass valorization to THFOL, THFOL applications, and the scope of this study. Here, FFA, FOL, 2-MTHF, 2-MF, CPO, and THP denote furfural, furfural alcohol, 2-methyltetrahydrofuran, 2-methyl furan, cyclopentanone and tetrahydropyran, respectively. Reproduced with permission from [189], Elsevier, 2021.

**Table 1 nanomaterials-13-02274-t001:** Sizes of NPs obtained in zeolites by different methods.

Preparation Method	Zeolite Type	NP Type	NP Size, nm	Ref.
Impregnation	ZSM-5	NiO	10–20	[118]
	HY	RuW	1–4	[106]
	Y	Ru	~1	[119]
	HZSM-5	Pd	3.5	[120]
	H-β	Ru	2.5	[114]
Mixing	Y	Co	14–16	[121]
	ZSM-5 and Y	Cu/Zn	6.2 ± 2.0	[111]
Ion exchange	BEA	Ni	3.5 or 6.1	[113]
	H-β	Ru	1.3	[114]
Encapsulation	HZSM-5	Co	3.0 ± 0.5	[103]
	HZSM-5	Cu	3.6	[103]
	Silicalite-1	Cu	1.8	[116]
	HZSM-5	Pd	2.1 ± 0.5	[71]
	TS-1	Cu	1.8	[117]
	HZSM-5	Ni	2–5	[122]

## Data Availability

Not applicable.

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
