# Peer review of "Design of Bifunctional Nanocatalysts Based on Zeolites for Biomass Processing"

_nanomaterials, 2023, doi:10.3390/nano13162274_

Round 1
Reviewer 1 Report (Previous Reviewer 3)
The modifications, carried out in the resubmitted manuscript, a little bit improved the scientific level of the revised one, though these changes did not bring essential improvements in the paper’s content. E.g. there are missing the authors own evaluations of the experimental results and/or the processes, emphasizing them in separate points e.g. in the conclusion. In a review such an evaluation may be essential. On the other hand, the manuscript can be accepted for publication in its presented form, as well
Author Response
Comment. The modifications, carried out in the resubmitted manuscript, a little bit improved the scientific level of the revised one, though these changes did not bring essential improvements in the paper’s content. E.g. there are missing the authors own evaluations of the experimental results and/or the processes, emphasizing them in separate points e.g. in the conclusion. In a review such an evaluation may be essential. On the other hand, the manuscript can be accepted for publication in its presented form, as well.
Response. We believe we expressed our opinion regarding literature examples throughout the revised manuscript and especially in Conclusion.
Reviewer 2 Report (New Reviewer)
Authors reviewed the design of catalysts consisting of mono- or bimetallic nanoparticles (NPs) and zeolite supports for biomass processing, and the synergy between metal sites and acid sites. This work fills a gap in the field by discussing the synergy between metal sites and acid sites from the perspective of NPs/zeolite catalyst design. It is considered as a novel and comprehensive review that meets journal requirements, but also needs to respond some questions. Some comments are as following.
1. Type of zeolite-As mentioned in the abstract, the main content of this review is to discuss the main methods of preparing NPs in zeolites, concentrating on the methods that allows the best interplay (synergy) between metal and acid sites. Therefore, authors should supplement to introduce the difference in acidity of different types of zeolites such as the differences in the ratio of Brønsted acid to Lewis acid.
2. Page 4, Line 118-119-The correlation between the NP/catalyst properties and the catalytic process are not discussed.
3. The effect of zeolite porosity is mentioned several times in the paper, but it is not mentioned in the abstract and summary.
4. Differences in NPS due to different preparation methods should be presented in a table.
5. Page 6, line 182-184 -“On the other hand, small, well distributed NPs……” How does this relate to ion exchange? It's not illogical. This should be modified. Moreover, I suggest authors to improve the language throughout the paper.
6. Page 8, line 252-256-The summary here is too general and it needs to be revised. For example, list the advantages of encapsulation such as limiting NPs growth and inhibition of NPs aggregation, leaching, etc.
7. Page 15, line 517-522-No useful information is given out from this example.
8. Page 16, line 559-560-This example assigns their success to the synergy between NPs and alkaline sites, which is inconsistent with the main idea of the review.
9. More new references should be cited, such as
1.Fuel, 2023, 340, 127567;
2.Fuel Processing Technology, 2023, 249, 107860;
3.ACS Catalysis, 2023, 13, 11, 7499-7513.
Minor editing of English language required
Author Response
Comment 1. Type of zeolite-As mentioned in the abstract, the main content of this review is to discuss the main methods of preparing NPs in zeolites, concentrating on the methods that allows the best interplay (synergy) between metal and acid sites. Therefore, authors should supplement to introduce the difference in acidity of different types of zeolites such as the differences in the ratio of Brønsted acid to Lewis acid.
Response 1. The requested information has been added to Section 2.
Comment 2. Page 4, Line 118-119-The correlation between the NP/catalyst properties and the catalytic process are not discussed.
Response 2. The information about the catalytic process has been added.
Comment 3. The effect of zeolite porosity is mentioned several times in the paper, but it is not mentioned in the abstract and summary.
Response 3. In the revised manuscript zeolite porosity has been mentioned in the Abstract and Summary.
Comment 4. Differences in NPS due to different preparation methods should be presented in a table.
Response 4. The table has been added to the revised manuscript.
Comment 5. Page 6, line 182-184 -“On the other hand, small, well distributed NPs……” How does this relate to ion exchange? It's not illogical. This should be modified. Moreover, I suggest authors to improve the language throughout the paper.
Response 5. The above paragraph has been modified. Moreover, the manuscript was proofread by the native English speaker (and a chemist).
Comment 6. Page 8, line 252-256-The summary here is too general and it needs to be revised. For example, list the advantages of encapsulation such as limiting NPs growth and inhibition of NPs aggregation, leaching, etc.
Response 6. The summary has been revised.
Comment 7. Page 15, line 517-522-No useful information is given out from this example.
Response 7. This paragraph has been rewritten.
Comment 8. Page 16, line 559-560-This example assigns their success to the synergy between NPs and alkaline sites, which is inconsistent with the main idea of the review.
Response 8. In the majority of cases, acid sites of zeolites participate in the catalytic reaction. That is why synergy between metal and acid sites is crucial. Nevertheless, in some cases basic sites or alkaline species (introduced for modification of zeolite) are beneficial for the catalytic reaction and their interactions with metal species are of interest. We do not see a conflict here.
Comment 9. More new references should be cited, such as
1.Fuel, 2023, 340, 127567;
2.Fuel Processing Technology, 2023, 249, 107860;
3.ACS Catalysis, 2023, 13, 11, 7499-7513.
Response 9. We thank the reviewer for suggested references, but unfortunately, none of them uses zeolites as NP supports. Thus, they are irrelevant to our review.
Reviewer 3 Report (New Reviewer)
1. The format of figures should keep consistent. For instance, Fig 2 use lowercase letters without bracket, while fig 9 use uppercase letters for numbering.
2. The title of the review is broader than the scope content in the main text. "bifunctional nanocatalysts" can refer to a lot of inorganic, organic or hybrid nanomaterials. The authors might consider to have something like "zeolite supported" in the title.
3. The introduction of "bifunctional" in line 48, 49 is a little bit unclear and confusing. The author may consider to put more details here. What is the difference or advantages compared to other heterogeneous catalysts.
4. The authors may consider to re-structure the main text. For example, the "Influence of porosity on mass transfer and catalytic processes (4.7)" part seems to talk about the zeolite porosity instead a modification method/approach. It should be moved to part 5. Or it could be translated to some method or approach to stay.
5. There are some cross overlap or uncleary between methods/approaches discussed in section 3 and 4. For example, what is the difference between 3.2 mixing and 4.4. post-fabrication modification. Can 4.6. Morphology modification be considered as part of 3.4. Encapsulation?
The quality of English language needs to be improved in terms of readability and professionalism. The authors should try to use shorter sentences to deliver the ideas. For example, the authors can shorten the sentence in line 493 by pre-defining the "direct biomass processing" earlier. Also, the sentence in line 239, 240, 528, 627, etc. can be shortened by minimizing the subordinate clause.
Author Response
Comment 1. The format of figures should keep consistent. For instance, Fig 2 use lowercase letters without bracket, while fig 9 use uppercase letters for numbering.
Response 1. Please note that in a review we do not prepare figures but only reproduce them from original research articles, as is indicated at the end of each Figure caption. That is why no changes can be made.
Comment 2. The title of the review is broader than the scope content in the main text. "bifunctional nanocatalysts" can refer to a lot of inorganic, organic or hybrid nanomaterials. The authors might consider to have something like "zeolite supported" in the title.
Response 2. The title has been revised.
Comment 3. The introduction of "bifunctional" in line 48, 49 is a little bit unclear and confusing. The author may consider to put more details here. What is the difference or advantages compared to other heterogeneous catalysts.
Response 3. An additional explanation has been provided for the term “bifunctional catalysts”. Their advantages compared to conventional catalysts have been emphasized.
Comment 4. The authors may consider to re-structure the main text. For example, the "Influence of porosity on mass transfer and catalytic processes (4.7)" part seems to talk about the zeolite porosity instead a modification method/approach. It should be moved to part 5. Or it could be translated to some method or approach to stay.
Response 4. When we define “modification” in the first paragraph of section 4, we indicate that it can be modification of zeolite during its formation. But we agree with the reviewer that some restructuring can be useful and placed a subsection on porosity at the beginning of Section 4.
Comment 5. There are some cross overlap or uncleary between methods/approaches discussed in section 3 and 4. For example, what is the difference between 3.2 mixing and 4.4. post-fabrication modification. Can 4.6. Morphology modification be considered as part of 3.4. Encapsulation?
Response 5. We believe these subsections are appropriate where they are. For example, in 3.2. and 4.4. we discuss the same paper, but different aspects. In the former case, it is a method of the NP fabrication in zeolite, while in the latter case, it is post-fabrication modification of the catalyst with bases. Subsection 4.6 (Morphology modification) discusses methods to introduce defects, small pores, or traps to control a NP size, while subsection 3.4. discusses an encapsulation method. The whole point of morphology modification is to form small NPs using simple impregnation, not necessarily encapsulation.
Comment 6. The quality of English language needs to be improved in terms of readability and professionalism. The authors should try to use shorter sentences to deliver the ideas. For example, the authors can shorten the sentence in line 493 by pre-defining the "direct biomass processing" earlier. Also, the sentence in line 239, 240, 528, 627, etc. can be shortened by minimizing the subordinate clause.
Response 6. We made changes indicated by the reviewer and polished the language throughout the manuscript. Moreover, the manuscript was proofread by the native English speaker (and a chemist).
This manuscript is a resubmission of an earlier submission. The following is a list of the peer review reports and author responses from that submission.
Round 1
Reviewer 1 Report
The review consists of an arbitrary enumeration of articles, presented without structure, logic and concept,following are the main concerns:
1. A major shortcoming of the article is that the main question to be addressed is not clearly defined.
2.The topic is not very original, and presented as it is, also not very relevant.
I don't think that the scientific community will benefit much from its publication.
no big problem with the English
Reviewer 2 Report
In this paper, the authors summarized recent advances of zeolite/NPs as catalysts for biomass processing. The review started with a brief introduction of biomass processing could be value-adding process and point out zeolite/NPs are promising candidates, followed by indroduction of zeolites, ways to introduce NPs in zeolites, modifications of zeolites, and the interaction between zeolite acid sites and NPs with abundant cases. In the following part, biomass processing reactions were briefly classified and discussed. Overall, this is a very ambitious review with a lot of state-of-the-art cases from recent years. However, the inspiration is not clearly stated, and the logic in some sections is not easy to understand. Thus, I suggest publishing it after major revision. Below are some of the detailed questions.
First, in the introduction part, Could the authors provide more information about the biomass types produced yearly, e.g. wet or dry, and how they could be processed individually? I don’t think zeolites with NPs is the ultimate solution for all cases, and it could be better to identify which cases they are suitable for. If possible, a quantitative analysis of the market and the requirement could make the big picture clearer and help the readers understand the most promising areas to explore.
In the writing of each section, it could be easier to understand if the different parts are classified and stated more clearly, and better organized. For example, in Section 3.2, the authors stated that SnO2 NPs could be incorporated in β zeolite, and the shifted to a mixed Pd/ZnO case where the tandem catalyst is better than a bifunctional catalyst. Next, several cases of bifunctional catalysts with synergistic effects were listed. I don't know why we shall care about SnO2 NPs in zeolite, and whether bifunctional catalysts are overall better or worse than tandem or other normal catalysts. The organization of information is not perfect although the authors did try their best to provide as much information as they could.
Although zeolite themselves have larger surface area and open channels, the size of the zeolites still affects the mass transport and thus, the overall efficiency. It would be great if influence from this aspect could be considered.
Reviewer 3 Report
Authors reviewed the preparation of bifunctional catalysts, using mono- or bimetallic nanoparticles incorporating them into zeolite support, with the aim its application for biomass (waste) processing. This manuscript summarizes the important results of the structural modification of the zeolite support to provide suitable stability and catalytic activity of the catalysts. The manuscript surveys the most often used methods for distribution of catalysts nanoparticles into support and then it discusses the modification methods of zeolite to increase its properties for integration of catalysts and its catalytic activity. Then the readers get brief information of the processing of biomass producing different products.
Generally, it can be stated that the information content of paper is acceptable, but the evaluation of the literature results seems to be missing. Accordingly, a few recommendations to be completing this manuscript:
- a brief evaluation of the methods at the end of the following points: paragraph 3.1-3.4. 4.1-4.6; this should involve, which methods can be regarded as the best in technological points of view;
- please rank the most often used direct and the indirect biomass processing emphasizing their efficiencies; it is not enough to show some examples of these two types of catalytic processing;
- please give a short looking out regarding the future technologies of the biomass processing;
- Please emphasize the most important results in the biomass processing in the conclusion;
I recommend to complete the content of this paper with the detailed analysis of the up to now achieved results in the catalytic processes of the biomass. This can essentially increase of the scientific level of this review, which is needed for publication of this manuscript.